# How to Predict Premature Multiphase Steel Cracks during Edge Flangeability

Lucas Salomao Peres [1,*], João Henrique C. Souza [2,*] and Gilmar Ferreira Batalha [1,*]

1   Department of Mechatronics and Mechanical Systems Engineering, Polytechnic School of Engineering, University of São Paulo, Av. Prof. Mello Moraes, 2231, Cidade Universitária, São Paulo 05508-900, SP, Brazil
2   Postgraduate Program of Mechanical Engineering, Engineering School, Federal University of Rio Grande, Av. Italia, km 8-Bairro Carreiros, Rio Grande 96203-900, RS, Brazil
*   Correspondence: lucassalomaoperes@gmail.com (L.S.P.); joaoh_cs@hotmail.com (J.H.C.S.); gfbatalh@usp.br (G.F.B.)

**Abstract:** The present paper makes a critical review based on the literature and presents examples of experiments developed by the authors, showing how the hole expansion ratio test (HERT) could be useful to understand and avoid premature cracks caused by flanging operations in sheet metal parts made of advanced high-strength steels. An approach based on damage theory was evaluated along with the mechanical tests necessary to understand the phenomenon, the influence of the trimming process, and the correlation between experimental mechanical testing and simulations. The procedures presented in this work allow for the prediction of edge cracks, often verified after flanging steps during the stamping process, allowing for reductions in tooling costs and setup loops.

**Keywords:** damage theory; edge cracking; dual phase steel; hole expansion test





## 1. Introduction

The automotive market seeks weight reduction, mostly based on reducing the thickness of the material and increasing its mechanical resistance, which is only possible by a comprehensive knowledge of the mechanical behavior of the materials [1]. This poses challenges to the stamping process, as higher levels of strength are accompanied by lower stretching limits. The first point is the reduction in the formability of the material during stretching; the second point is the influence of high strengths on deep drawing; the third is the trimming process; and the fourth is the flangeability, especially in multiphase materials prone to premature edge cracks. Premature edge cracking is currently the most challenging issue for stamping specialists, and several research works aim to develop more reliable methodologies for its prediction [2]. In this paper, premature cracking caused by edge flanging in high-strength steels is evaluated by a literature review and examples of experiments developed by the authors.

When comparing resistance versus elongation in steel grade, in Figure 1, the most resistant materials with the capacity of good formability are in the middle of the diagram, characterized by ferrite–bainite, TRIP, complex phase, dual phase, and third generation steels, all of which are composed of more than one phase: ferrite plus martensite, austenite or bainite, or all of them. TRIP and third generation steels have another factor, deformation-induced phase transformation, from austenite to martensite [3]. Crack propagation in ductile behavior is a process to reinitiate a crack after it has occurred. For a flat body with a similar microstructure, it is possible to have different crack tip directions independent of preexisting cracks in the region [4].

Phase transformation, as it occurs in third generation and TRIP steel, is another complexity factor to the crack expansion phenomenon; during this transformation, the material can absorb energy, deviating the crack tip. This can sometimes influence the

hole expansion results, but not always positively. When compared to other advanced high-strength steels (AHSS), it is not possible to have a direct correlation; some TRIP materials can have a worse performance than dual-phase materials with a superior mechanical strength [5].

It is easier to understand the phenomenon of premature cracking using dual-phase 980 steels (DP980). This material is composed of martensite and ferrite, and it is possible to observe the interaction of these phases in hole expansion and the reason why it is different from monophase materials.

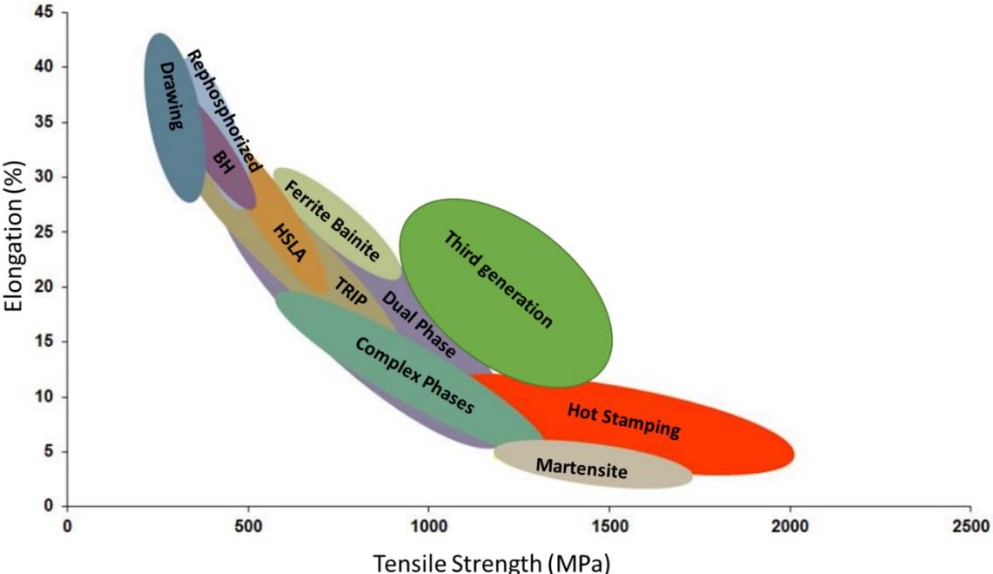

**Figure 1.** Relation between steel grades' elongation vs. resistance (adapted from KEELER et al., 2017 [6]).

Moreover, finite element analysis (FEA) based solely on traditional mechanical input for AHSSs is not capable of predicting cracks. The texture of the material and the size of the grain are directly related to the crack mechanisms in the expansion of the hole [7].

The flangeability is an intrinsic characteristic of the material and is the ability of the material to increase or decrease the perimeter of its edge during the drawing process. During stamping, it is possible for the same part to increase and decrease its perimeter edge, increasing the complexity of the evaluation and its relation to the trimming process quality [8].

## 2. Damage Theories

These theories are based on understanding that no material is perfectly continuous. According to Griffith, the deformation of voids inside the materials increases the inner tension inside the crystalline matrix. This increase in the tension can also be influenced by nucleation, coalescence, and void growth. Figure 2 shows a schematic of entrapped voids that grow by the action of the tensile lines [9]. The evaluation methods using the models of ductile fractures and plastic instability as criteria were proposed by Yoshida H. et al. (2013) [10].

According to Gross and Seeling [11], damage can be initiated by the nucleation, growth, and coalescence of micro-voids. Grain boundaries and different phases or inclusions can be a barrier to the dislocation movement, resulting in the nucleation of a crack. Thus, microscopic plastic flow or micro-void density generated by punching is not the dominant factor in the stretch–flange –formability of UHSS sheets.

The damage theory assumes that the voids are far from each other and do not interact. The elastic part is dominated by Hooke's law, and the plastic part is dominated by the yield condition. Anderson [12], based on the observations, says that void nucleation is faster under high triaxiality stresses. Void nucleation in large particles is facilitated by increasing the number of defects that cause crack initiations.

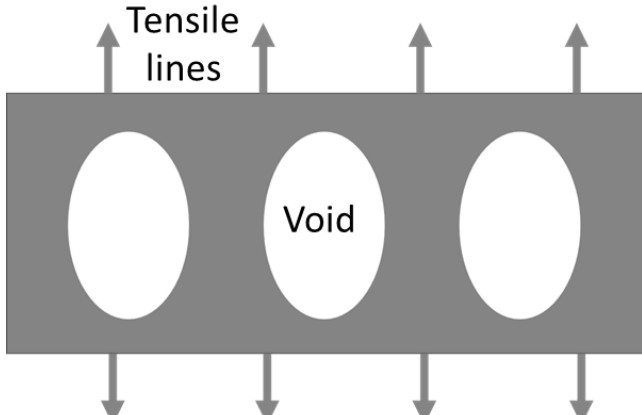

**Figure 2.** Tensile lines that influence the growth of voids (adapted from Anderson, 2017 [12]).

Orowan [13] explained that Griffith's theory for brittle-behavior solids could not be applied to ductile-behavior materials. The same author observed that laboratory tests showed that cleavage in a steel fracture can increase at slow deformation rates, different from brittle materials that grow fast. There is significant plastic deformation due to the plastic constraints that result in fractures by cleavage. Steel fractures may begin with ductile cracking followed by considerable plastic deformation. Orowan could not adjust Griffith's theory to include plasticity. Years later, based on Orowan's theory, Irwin [14] improved existing theories. Irwin observed extensive plastic deformations in X-ray tests, highlighting the flaw in Griffith's theory.

Irwin (1961) [14] structured the analysis using stored strain energy, surface energy, and the work for plastic deformation. The analysis was based on the energy balance defined by Griffith.

Based on that principle, Cockcroft–Latham [15] formulated the theory of maximum principal stress, using multiaxial fractures related to crack flow. Cockcroft–Latham (1968) [15] proposed the energy fracture criterion, which states that fractures depend on the integral of the principal tensile stress. Thus, for a given material, this criterion suggests that fractures occur when the integral of the tensile stress reaches the critical value.

Later, the formula was modified by Oh, Chen, and Kobayashi [16], proposing that it is possible to include a material factor that influences the mechanics of damage in the drawing process. $D_i$ is a constant representing the workability of the material (1). The relationship between the material and the microstructure is expressed in Equation (1) by $C_{val}$; $\varepsilon_p^{-f}$ is the equivalent plastic strain in a fracture, $\overline{\varepsilon_p}$ is an equivalent plastic strain, and $\sigma$ is the stress.

$$D\iota = \frac{1}{C_{val}} \int_0^{\varepsilon_p^{-f}} f(\sigma)d\overline{\varepsilon_p} \tag{1}$$

Lee (2005) [17] compares seven fracture criteria:

- Bao–Wierzbicki model: best adapted to most mechanical tests due to Lee's calibration in his study;
- Maximum shear stress: similar to Tresca's yield condition, but using the critical shear stress, not the shear flow stress;
- Cockcroft–Lathan: as explained before, best captured the process of crack formation;
- Wilkins's model: also based on the critical value and on the critical dimension, which should be calibrated for each designated test;
- Johnson–Cook fracture criterion: postulates that fractures are a function of stress triaxiality, temperature, and strain rate, but is good to express the nonconstant triaxiality loading process;
- Constant equivalent strain fracture criterion: to reach the plastic strain reaches a critical value for fractures;

- The fracture forming limit diagram: based on the relation between the maximum and the minimum stress, and it is highly used in stamping processes to understand critical zones for tooling processes, but it is however inappropriate to give information on the crack beginning [17].

### 3. Mechanical Test

There are different mechanical tests, and the most used test to define steel properties is the tensile test. Figure 3 represents some results for DP980 obtained by the author following ISO 6892-1 [18].

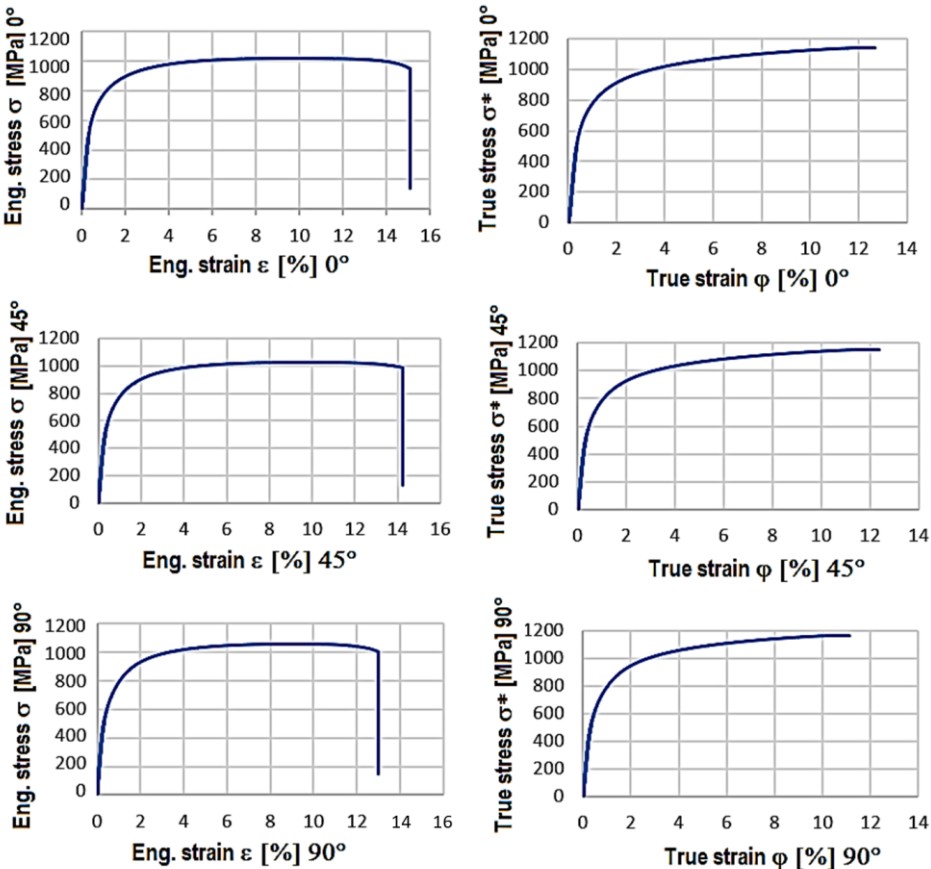

**Figure 3.** Stress–strain curves: DP980 tensile tests at 0, 45, and 90° (by author).

This test could provide a basis for FEA analysis, but it is not enough; there are two tests derived from the tensile test with equal relevance: the plastic strain ratio (r) test [19] and the tensile strain-hardening exponents [20] test. These results quantify the capacity of the material to reduce its thickness during the tensile test and the hardening capacity of the material during the tensile test for homogeneous elongation, respectively.

Another possibility is a forming limit diagram (FLD) [21]. An FLD is a graphical representation of the limits to forming, i.e., the major and minor stresses where local necking occurs (Figure 4). Different specimen geometries generate different deformation patterns, as expressed by the Nakajima test (Figure 5) [22].

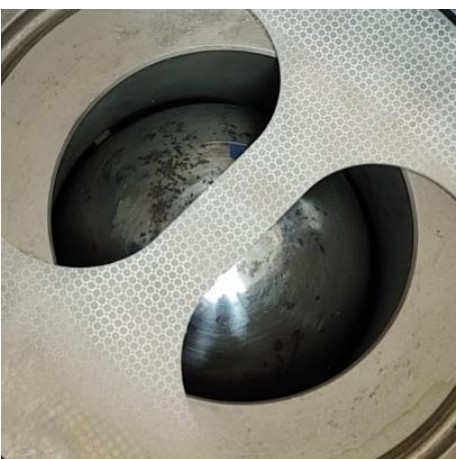

**Figure 4.** Nakajima test during deformation (source: author).

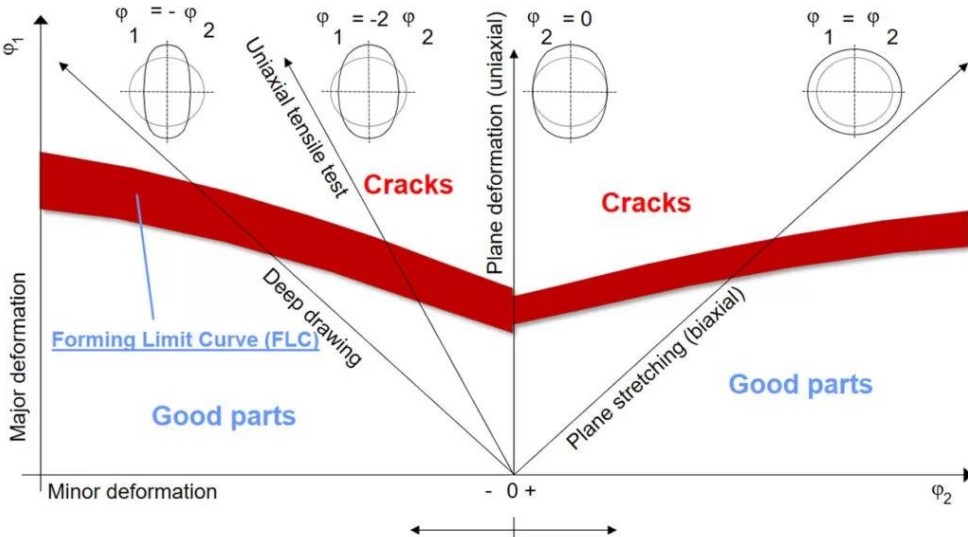

**Figure 5.** Nakajima probe dimensions and forming limit diagram test (Reproduced with permission from Souza, J.H.C.; published by FormingWorld, 2022 [23]).

Another example is the cup test by Erichsen. Although very simple, it is sometimes still used to understand the maximum capacity of the equiaxial deformation up to the rupture and to compare the ductility of different materials [24]. Finally, there is the hole expansion ratio test (HERT). This test is important to compare the flangeability of sheet materials. Its characteristics are discussed in Section 5.

## 4. Trimming

The quality of the trimming process in stamping processes must be constantly evaluated regarding the sheared edge morphology since it represents one of the greatest influencing factors on edge flangeability during the drawing process. The goal of stamping processes is to provide a part with the correct geometry, required surface quality (variables depending on application), and desired features (e.g., holes of the correct size and location, properly oriented with no splits or objectionable wrinkles).

The latter two, tearing (splits) and wrinkles, are strongly influenced by the selection of material. The trimming process occurs before hole expansion is standardized; there are thus no considerable differences between two materials of a similar resistance and elongation. In this part, the texture does not prevail over the resistance [25]. The quality of trimming is a function of the clearance and the resistance; when clearance decreases and resistance increases, burr and rollover are reduced, but the wear of the tool increases.

Four zones can be observed: rollover, burnish, fracture, and burr. When the resistance increases, it is normal to see the alternation of burnish, caused by shear stress and fractures, as shown in Figure 6.

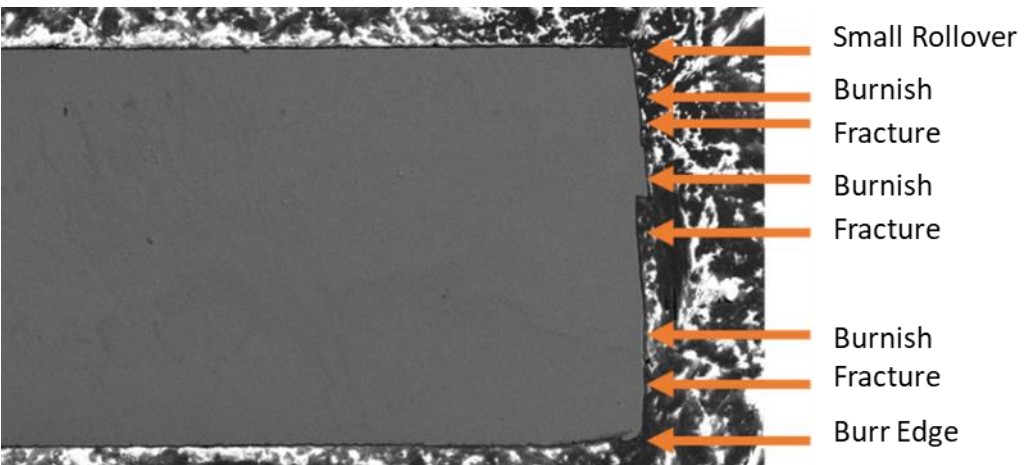

**Figure 6.** Trimming characteristic of the DP980 punched hole (source: author).

Different hole finishing preparation methods (e.g., punching, milling, and laser cutting) can increase the damage levels causing crack initiation at the hole edge. Maximum damage is normally caused by conventional punching, followed by milling, and later by lasers. All these processes can cause small cracks and initiate a fracture [25]. The stretch flangeability of an ultra-high-strength steel sheet with a small ductility can be increased by improving the quality of the sheared edge [25].

Warm and hot punching using resistance heating was developed to improve the quality of sheared edges of an ultra-high-strength steel sheet. As the heating temperature increased, the depth of the shiny burnished surface on the sheared edge increased and that of the rough fracture surface decreased. The rollover depth and burr height of the sheared edge exceeded 800 °C. Although the roughness of the burnished surface was almost constant, the roughness of the fracture surface increased by 650 °C. The local resistance heating of the shearing region was efficient for warm and hot shearing. Warm and hot shearing of ultra-high-strength steel sheets has been found to be effective for improving the quality of the sheared edge and in reducing the shearing load. Since deep drawing has problems such as fractures, seizures, and tool wear due to large deformations, bending is a preferable process for ultra-high-strength steel [25].

## 5. Hole Expansion Ratio Test (HERT)

The stretch flangeability was evaluated using the hole expansion test, and the punched hole in the sheet was expanded with a conical punch with an angle of 60°. In the hole expansion, the sheared edge was expanded uniformly in the hoop direction so that the tensile stress acted on the edge. The burr of the sheet was set without a touch of the punch on the opposite side [26].

The first part consists of the cutting process and the hole expansion test based on standard ISO 16630 [27]. For the trimming and hole expansion test, a universal sheet metal testing machine was used, model ZwickRoell BUP 600, with a hole expansion test tool kit (Figure 7). After trimming, the sample was placed with the burr edge up. The test was stopped when a visible crack started. Founded on the standard in Equation (2), the diameter was measured before the test ($D_0$) and after the test ($D_e$).

$$\lambda = \frac{D_e - D_0}{D_0} \times 100 \qquad (2)$$

According to Paul (2018) [28], it is not possible to compare the tensile test by uniaxial tests with the hole expansion, but the value of the true stress–strain is necessary to calculate using finite element analysis in the stamping process.

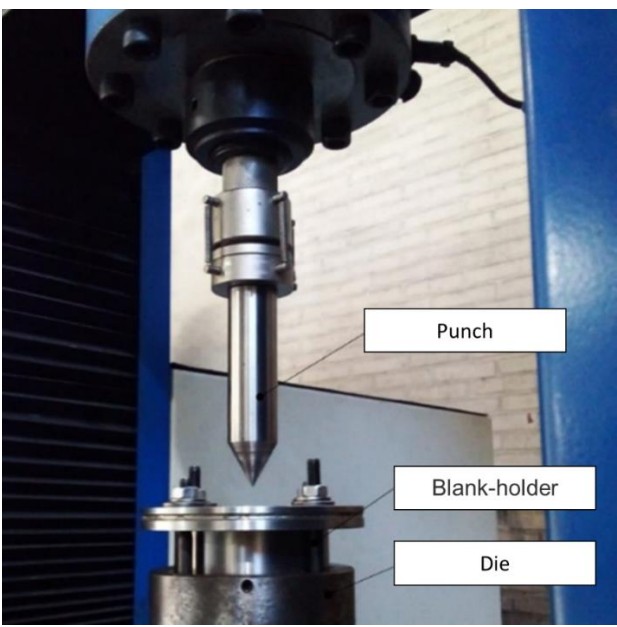

**Figure 7.** Hole expansion test (Reproduced with permission from Rosiak, A. et al.; published by 41st SENAFOR, 2022 [29]).

Figure 8 shows the regions in which the cracks started almost simultaneously. Since hole expansion is normally not automatized, human assistance was necessary to stop the equipment.

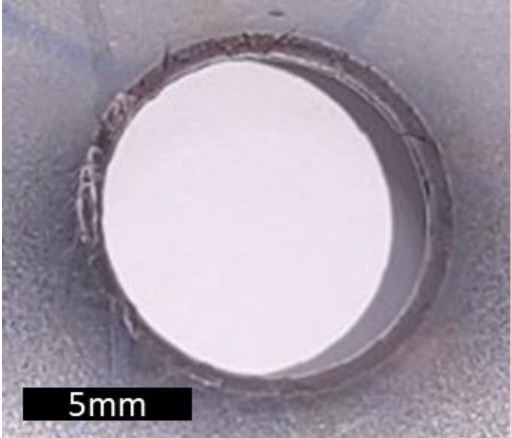

**Figure 8.** Sample after the hole expansion test (source: author).

## 6. Microstructure

Raabe (2020) [5] conducted a very detailed study on advanced high-strength steels and exposed the relation between the microstructure and hole expansion. He presented a graph containing the relation of multiphase steel elongation, oppositely proportional to the hole expansion performance. This aids in the understanding of why the present study focuses on the crack area (Figure 9) to identify void growth [5].

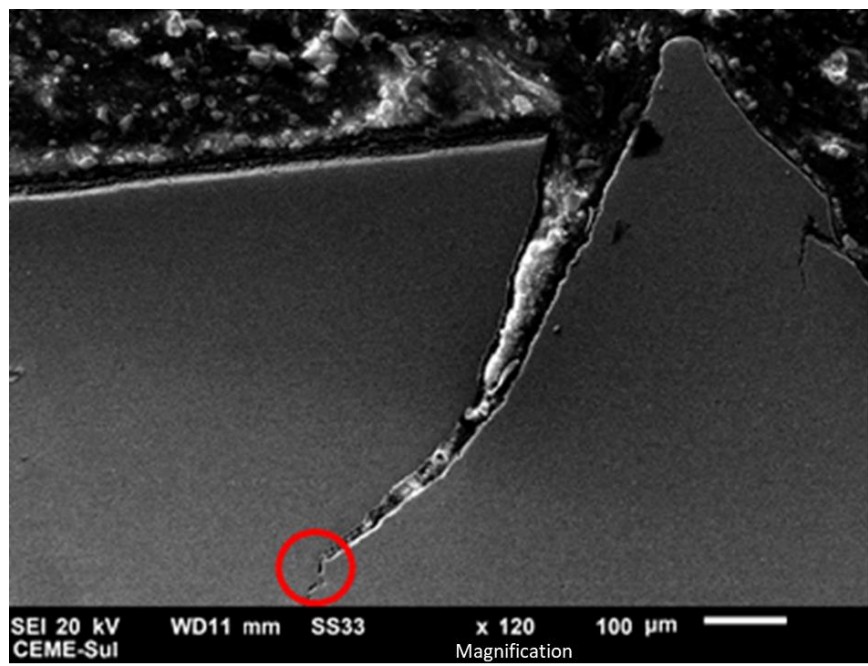

**Figure 9.** SEM of the material in the layer: crack region (source: author).

Figure 10 allows for the observation of the formation and growth of a void around the martensite. As Kim (2017) explained, it is a decohesion interface; the relationship between the grain boundary of martensite and ferrite can create a void [30].

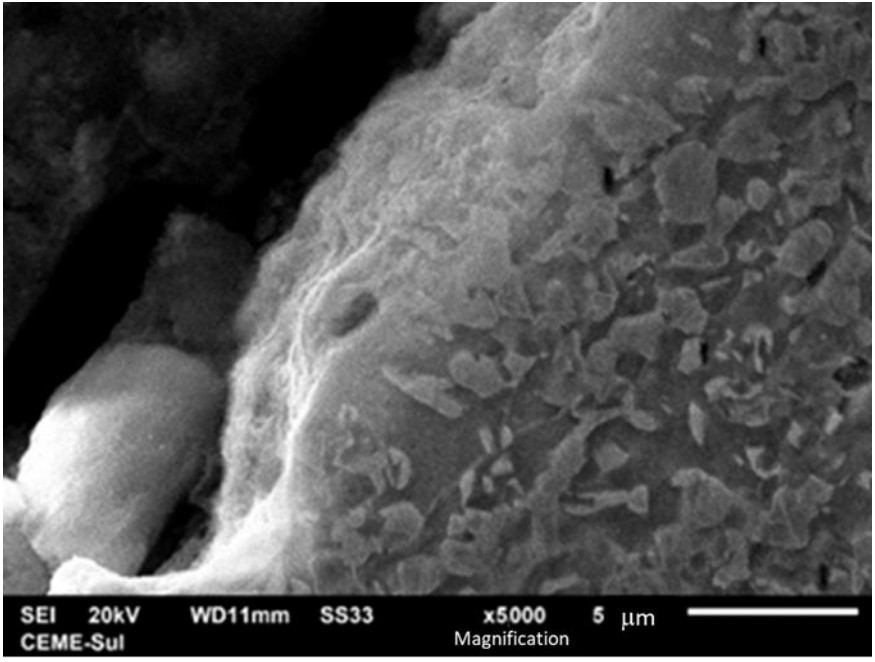

**Figure 10.** Formation of voids around the martensite phase (source: author).

The void continued growing (Figure 11) until a barrier was found; this barrier could be another grain of martensite. To understand this phenomenon, Zhang (2019) tested the relationship between the texture of different reduction rates and how this could influence the material resistance [31].

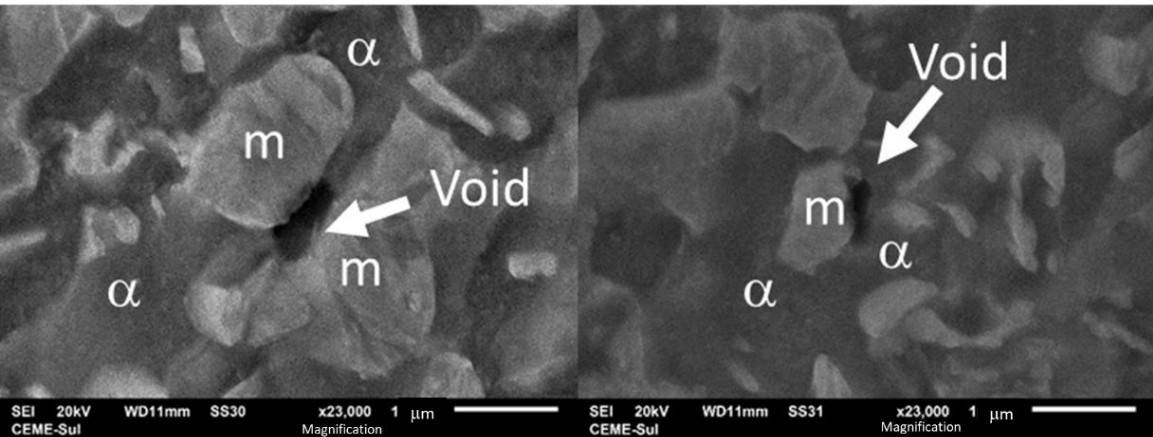

**Figure 11.** Void growth between the ferrite ($\alpha$) and martensite (m) (source: author).

The relation between premature cracking and flangeability was discussed by Akela, Kumar, and Bakachndran (2021), comparing two equivalent tensile strengths, DP 600 and HS 700 (high-strength low alloy). In their observations, the materials with a more uniform distribution and a finer-grained ferritic matrix had a greater hole expansion ratio compared to HS 700 [32].

## 7. Correlation between Experimental Mechanical Testing and Simulations

To replicate the test in a digital twin, it was necessary to recreate the intrinsic and extrinsic conditions for the material and the tool and the mechanical properties, such as flow stress, Young's module, Poisson's ratio, Hill's quadratic median anisotropy for the three directions [19], and density [33]. Figure 12 shows a computed plot example of the principal stress $\sigma_1$ obtained after the FEM analysis carried out using the QForm FEM software.

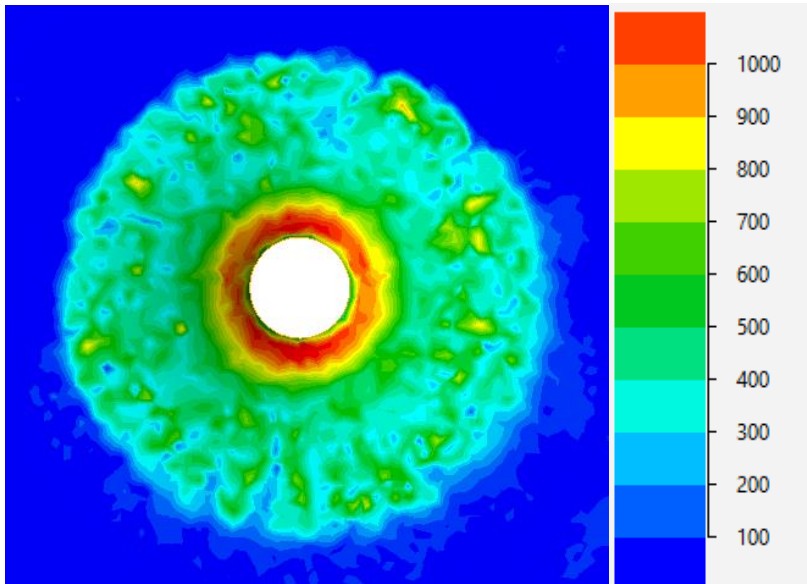

**Figure 12.** Computed plot of the principal stress $\sigma_1$ (analysis in QForm) (source: author).

The hole expansion test was used before to calibrate the methodology of Bao–Wierzbicki, stress-triaxiality dependent. This process allowed for the determination of the ductile damage and the progressive damage of aluminum 1050–H14 (monophasic). This was a good example of using a hybrid method to calibrate the parameters to introduce in the damage

model. This exposed the need to identify a damage model that can incorporate into FEA to predict the metal-forming process cracks [34].

There are different methods to calculate the damage criteria and evaluate damage; the method chosen for the present paper was the damage model of Cockcroft–Latham. According to the literature, this method is very accurate and is easily implemented to evaluate the workability of the material. The Cockcroft–Latham criteria relates plastic deformation and tensile stress, observing that voids grow by deformation [35]. There are many other ways to determine the damage module such as the principle of equivalence in deformation; this principle is based on thermodynamics and expresses the possibility of understanding the microstructure applied to rheological dislocations [36].

This is important to improve tool geometry, calculate the thickness distribution, and optimize the material flow to avoid premature failures. Some studies show that using a combination of practical and theoretical models to improve quality reduces development time and costs [37,38].

## 8. Conclusions

It is possible to understand why premature cracking occurs in multiphase steel: it derives from the decohesion interface between ferrite and martensite. However, there is still a long way to go to understand the relationship between the advanced high-strength steel microstructure and its flangeability.

In this work, it is proposed to follow the steps shown in Figure 13 to predict premature cracks and reduce costs and tooling setup time. However, it is necessary to follow the edge quality of the material during tool life in order to obtain satisfactory results. The hole expansion test is still very manual and depends on the accuracy of the operator and on experience to observe the start of the crack.

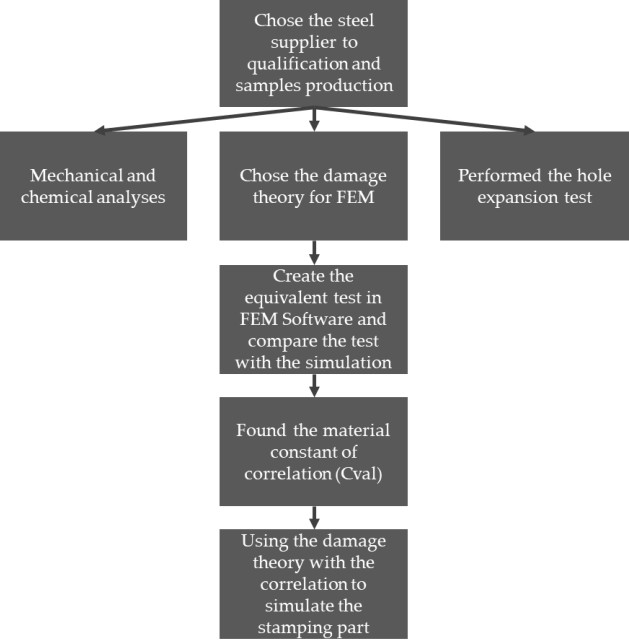

**Figure 13.** Methodology: schematic steps to predict premature cracking (source: author).

Although ultra-high-strength steel sheets seem promising for reducing the weight of automobiles, the stamping process is still challenging. The rise in strength leads to the increase in forming loads, spring back, and tool wear and the decrease in formability. To overcome these difficulties, improvements not only in material properties, but also in forming processes are required.

For the next step, it is possible to correlate the Cockcroft–Lathan damage theory with the hole expansion test and to determine the material intrinsic constant of the equation to

calibrate the model, allowing for the prediction of premature cracks in the stamping process. It is nevertheless necessary to understand the relationships between size and distribution of the grain phases and how they interact by virtual simulations. It must be possible to extract this information from a simple microscope, reducing time and the complexity of the process. The perspective of determining this correlation is very promising. The challenge is to automatize this process and to reduce the necessity of the specialization of the operator during simulations using FEA analyses.

**Author Contributions:** Conceptualization, L.S.P., J.H.C.S. and G.F.B.; methodology, L.S.P.; software, L.S.P.; validation, L.S.P., J.H.C.S. and G.F.B.; formal analysis, L.S.P., J.H.C.S. and G.F.B.; investigation, L.S.P., J.H.C.S. and G.F.B.; resources, L.S.P. and J.H.C.S.; data curation, L.S.P., J.H.C.S. and G.F.B.; writing—original draft preparation, L.S.P., J.H.C.S. and G.F.B.; writing—review and editing, L.S.P. and G.F.B.; visualization, L.S.P., J.H.C.S. and G.F.B.; supervision, G.F.B.; project administration, L.S.P.; funding acquisition, L.S.P. All authors have read and agreed to the published version of the manuscript.

**Funding:** This research received no external funding.

**Institutional Review Board Statement:** Not applicable.

**Informed Consent Statement:** Not applicable.

**Data Availability Statement:** Not applicable.

**Conflicts of Interest:** The authors declare no conflict of interest.

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
