# Peer review of "How to Predict Premature Multiphase Steel Cracks during Edge Flangeability"

_2673-4117, doi:10.3390/eng3040034_

Round 1

Reviewer 1 Report

The content of the paper does not reflect the impression that the title of the gives. 

The authors have not done sufficient literature survey or critical analysis to call their paper "critical review"

In some figures the caption contains (by author). If the figures are drawn by the authors, and are experimental output of the authors, then the authors have to mention how/using which data they obtained the figures. If they taken the image from some reference, they have to cite the reference.

In addition to problems in English language/grammar the text reflects the lack of rigour in the authors work.

Author Response

The article continues as a review, but the English text are fully corrected. The figures are reedited, and the bibliography is adjusted.

Reviewer 2 Report

Review’s comment

Title :How to predict premature multiphase steel crack during edge flangeability – A critical review

This manuscript show a critical review based on the bibliographic background and examples of experiments developed by the authors. It seems that how the hole expansion ratio test (HERT) could be useful to understand and avoid the premature crack caused by the flangeability of the advanced high strength steels. This paper describes the basic approach of edge flangeability and contents is easy to understand using common sense. I didn’t comment about detail context. But, I recommend to revise some errors and captions in manuscript. Please refer to the below comments.

In Figure 2, what is the meaning of the arrows and blue box with white hole? You should identify the parameters which you want to describe.

In line 106, ‘The Di is a constant representing the workability of the material in eq.(1)’ But, this parameter isn’t not present in eq.(1). Instead, the indicate that the DL means.

In lines 111-116, it will be helpful in review with the explained fracture models (seven fracture criteria including Bao-Wierzbicki Model, Wilkins Model, Cockcroft-Lathan ..).

In figure 7 (Trimming characteristic of the DP 980 punched hole (by author).), the change of direction (upper and bottom) will be better, in common, the rollover is upper. In addition the tool of punching should be shown with schematics.

In eq (2), the meaning of parameters  should be indicated in manuscript.

In line 215-216, the This is possible to observe in Figure 8 the formation and growth of a void around the 215 martensite. As Kim (2017) explained that it is a decohesion interface, the relationship be- 216 tween the grain boundary of martensite and ferrite can create a void (figure 10).

Figure 8 should be changed to Figure 10, right?

In line 234-235, Figure 10 shows a computed plot example of the principal stress 234 s1, obtained after FEM analysis carried out using the QForm FEM software.

Figure 10 should be charged to Figure 12, right?

Author Response

n Figure 2, what is the meaning of the arrows and blue box with white hole? You should identify the parameters which you want to describe. - Identified.

In line 106, ‘The Di is a constant representing the workability of the material in eq.(1)’ But, this parameter isn’t not present in eq.(1). Instead, the indicate that the DL means. - Ajusted.

In lines 111-116, it will be helpful in review with the explained fracture models (seven fracture criteria including Bao-Wierzbicki Model, Wilkins Model, Cockcroft-Lathan ..). - I try to rewrite to better understand.

In figure 7 (Trimming characteristic of the DP 980 punched hole (by author).), the change of direction (upper and bottom) will be better, in common, the rollover is upper. In addition the tool of punching should be shown with schematics. - Chenged the direction.

In eq (2), the meaning of parameters  should be indicated in manuscript. - It was indicated.

In line 215-216, the This is possible to observe in Figure 8 the formation and growth of a void around the 215 martensite. As Kim (2017) explained that it is a decohesion interface, the relationship be- 216 tween the grain boundary of martensite and ferrite can create a void (figure 10).

Figure 8 should be changed to Figure 10, right? - Corrected

In line 234-235, Figure 10 shows a computed plot example of the principal stress 234 s1, obtained after FEM analysis carried out using the QForm FEM software.

Figure 10 should be charged to Figure 12, right? - Corrected;

Reviewer 3 Report

Dear authors, I am sending comments on your work in the file.

Author Response

-why is the next section numbered 1 Mechanical test? - Corrected
There is no information on whether the DP980 material was tested by the authors during the tensile tests?  - Ajusted
On what equipment were the tests carried out and what parameters were used (fig.4)?- Included
There is no information about the test run for the Nakajima method (fig.5)?-  Included
The Trimmingnie section has no numeration? - included
There is no information in the text whether Fig.7 is from the authors' own research? - Included
In lines 163-166 I noticed the background. Please correct.
Why is the next section number 5 (Hole Expansion Ratio Test)? - ajusted
Please correct degrees in line 184
The results of the diameter measurement according to equation (92) are absent. - ajusted
Error in numbering of drawings. Twice there is a figure 8 - ajusted
Please indicate in the figure the phenomena occurring in the Hole expansion test (Figure Sample after the hole expansion).
Error in numbering of drawings. Please correct. - ajusted

Are the SEM images of the authors? No information about the type of material in the SEM images. No information after which tests they were taken.
The microstructure analysis is too general. Please provide a richer analysis based on the literature. - Included
The authors write (lines 231-235):
To replicate the test in a digital twin, it was necessary to recreate the intrinsic and extrinsic conditions for the material and the tool, mechanical properties such as flow stress, young module, Poisson ratio, Hill quadratic median anisotropy for the three directions [18], density [30]. Figure 10 shows a computed plot example of the principal stress, obtained after FEM 
analysis carried out using the QForm FEM softwar
Comment:
Please disclose in your paper the parameter values of mechanical properties such as flow stress, young modulus, Poisson ratio, Hill quadratic median anisotropy for the three directions, density.
Please plot the results of laboratory and numerical tests on a graph for comparison. included on figure
Conclusion
The conclusions presented can be supplemented by your own research.
Evaluation of work:
The work in its current form cannot be published:
- correct the numbering of the figures - ajusted
- correct the text as a whole + add your research findings.  - ajusted
The paper will be of value in relation to fracture mechanics.

Reviewer 4 Report

The authors provided an insightful critical review and recommendations on how to comprehend and avoid the premature cracking caused by the flangeability of advanced high strength steels.

Furthermore, the authors recommend which damage theory and mechanical tests are required to comprehend the phenomenon, as well as the impact of the trimming process and the correlation between experimental mechanical testing and simulation. The work is very well-conceived, with no flaws that should be addressed.

Author Response

Please see the attachment with the correction.

Reviewer 5 Report

The present work summarized the methods to predict premature crack during edge flagneability in multiphase steels. This is important for engineering. However, before pubilication there are some apects that were needed to be improved, as following:

1. It is not very clear that how to predict premature crack, even then author gave several testing method, but what relationship between them with premature cracking should be given, and how to predict by using these properties obtained by those tests? I strongly recommend to draw a schemical diagram to present this aspect.

2. It is better to add a section for giving perspective and challenges in the prection of prematrure cracking. A high-level review often gives important information to lead the future research topics or arise the hotpot on some aspects in field of prection of prematrure cracking.

Author Response

1. It is not very clear that how to predict premature crack, even then author gave several testing method, but what relationship between them with premature cracking should be given, and how to predict by using these properties obtained by those tests? I strongly recommend to draw a schemical diagram to present this aspect. - included on conclusions

2. It is better to add a section for giving perspective and challenges in the prection of prematrure cracking. A high-level review often gives important information to lead the future research topics or arise the hotpot on some aspects in field of prection of prematrure cracking. - improved the text to be more clear this topic.

Round 2

Reviewer 1 Report

The authors have not addressed any of my comments.

Author Response

Dear Reviewer,

Follow your requests (in italics):

The content of the paper does not reflect the impression that the title of the gives.  - "critical review" was removed from the title.

The authors have not done sufficient literature survey or critical analysis to call their paper "critical review" - From the first time the literature was increased, but there are limitation from the pages in the text, that limit the number of literature.

In some figures the caption contains (by author). If the figures are drawn by the authors, and are experimental output of the authors, then the authors have to mention how/using which data they obtained the figures. If they taken the image from some reference, they have to cite the reference. – Some of the figures was picture from my analyses in laboratory not published, following the methodology of the standards cited in the text. That images were inserted in the text to improve the capacity of the lectors to understand the text and the references. All others figures have references.

In addition to problems in English language/grammar the text reflects the lack of rigour in the authors work. – All the text was analyzed by a specialist and corrected.

Best Regards

Reviewer 3 Report

Dear Authors,

I accept the amendments made. 

Author Response

Thank you

Round 3

Reviewer 1 Report

The authors have incorporated my suggestions reasonably well.

They should do a spell-check once more before submitting. For e.g.: In the Introduction section on line 46 TRIP steel is written as tryp.

Author Response

Dear Reviewer

Thank you for your review, the text was fully spell-check and corrected.

Thank you very much for your advicess wich enriched the text.

Follow the corrected text atached.

Best regards.
